

# Accelerating the discovery of rare tree species in Amazonian forests: integrating long monitoring tree plot data with metabolomics and phylogenetics for the description of a new species in the hyperdiverse genus *Inga* Mill

Juan Ernesto Guevara Andino[1], Consuelo Hernández[2], Renato Valencia[3], Dale Forrister[4,5] and María-José Endara[1]

[1] Grupo de Investigación en Biodiversidad, Medio Ambiente y Salud-BIOMAS, Universidad de las Americas, Quito, Ecuador
[2] Laboratorio de Ecología de Plantas. Herbario QCA, Escuela de Ciencias Biológicas, Pontificia Universidad Católica del Ecuador, Quito, Ecuador
[3] Laboratorio de Ecología de Plantas. Herbario QCA, Escuela de Ciencias Biológicas,, Pontificia Universidad Católica del Ecuador, Quito, Ecuador
[4] Department of Biology, University of Utah, Salt Lake City, United States of America
[5] Estación de Biodiversidad Tiputini, Colegio de Ciencias Biológicas y Ambientales, Universidad San Francisco de Quito—USFQ, Quito, Ecuador

## ABSTRACT

In species-rich regions and highly speciose genera, the need for species identification and taxonomic recognition has led to the development of emergent technologies. Here, we combine long-term plot data with untargeted metabolomics, and morphological and phylogenetic data to describe a new rare species in the hyperdiverse genus of trees *Inga* Mill. Our combined data show that *Inga coleyana* is a new lineage splitting from their closest relatives *I. coruscans* and *I. cylindrica*. Moreover, analyses of the chemical defensive profile demonstrate that *I. coleyana* has a very distinctive chemistry from their closest relatives, with *I. coleyana* having a chemistry based on saponins and *I. cylindrica* and *I. coruscans* producing a series of dihydroflavonols in addition to saponins. Finally, data from our network of plots suggest that *I. coleyana* is a rare and probably endemic taxon in the hyper-diverse genus *Inga*. Thus, the synergy produced by different approaches, such as long-term plot data and metabolomics, could accelerate taxonomic recognition in challenging tropical biomes.

# INTRODUCTION

Amazon forests are among the most biodiverse ecosystems on earth. Recent evidence suggests that they harbor approximately 16,000 tree species (*Ter Steege et al., 2019*). However, documenting this extraordinary diversity has proved a daunting task for

Corresponding authors
Juan Ernesto Guevara Andino,
juan.guevara@udla.edu.ec
María-José Endara,
maria.endara.burbano@udla.edu.ec

ecologists and taxonomists. Since the first botanical explorations in the early 1700's the formal recognition of new Amazonian tree species has been characterized by a slow pace of species identification and poor taxonomy (*Ter Steege et al., 2016*). With more than 62% of the Amazonian tree flora occurring at very low densities (collectively compromising 0.12% of trees in the Amazon, *Ter Steege et al., 2013*), finding individuals with flowers and fruits that would allow definition and species delimitation becomes a challenge (*Baker et al., 2017*). In addition, tropical forests harbor a large number of congeneric species that are morphologically similar. For these group of species, morphological taxonomy and DNA barcoding are problematic for species identification (*Dexter, Pennington & Cunningham, 2010*; *Endara et al., 2018*). Thus, the challenge of finding and describing new species in Amazon forests is enormous (*Baker et al., 2017*; *Ter Steege et al., 2019*).

For biodiversity assessments and the development of conservation strategies, the implementation of accurate and fast species identification pipelines within and among sites is fundamental. Emergent technologies, such as untargeted metabolomics, can accelerate the discovery of species by identifying entities that may potentially be new to science. Specifically, defense-related chemical compounds, in combination with molecular networking (*Wang et al., 2016*), can create unique species-level fingerprints (also known as "chemocoding", *Endara et al., 2018*). This method can provide an additional molecular tool for species identification in unfertile material, confusing species, or in highly speciose genera where morphological distinctions between species are difficult and cannot be resolved using DNA barcodes. For these groups of plants, sequencing of many genes might be necessary in order to discriminate among species, and usually, only few individuals per species are sequenced. This can be time-consuming and expensive. Alternatively, with chemocoding, classification of thousands of samples simultaneously can be achieved in a fast, inexpensive and accurate way (*Endara et al., 2018*). This approach can be valuable for networks of forest plots since it can consistently help to differentiate species across sites and geographic ranges (*Endara et al., 2018*; *Guevara Andino et al., 2019*). In long-term monitoring plots, the regular study and collection can increase the chance of finding fertile individuals of rare or previously unknown specimens. In addition, working with distributed plots facilitates obtaining information on the distribution and abundance of specimens, which is essential for studies of ecology and conservation of new taxa (*Baker et al., 2017*). Thus, centering our chemocoding in long-term monitoring plots can greatly help increase the pace of new species recognition.

Here we show how the synergistic effect of long-term plot monitoring with chemocoding has helped to disentangle cryptic diversity in the hyperdiverse tree genus *Inga*. The genus *Inga* is one of the most abundant and speciose group of trees (>300 species) in Neotropical rainforests and has explosively radiated in the last 4–8 million years (*Pennington, 1997*; *Richardson et al., 2001*; *Lewis et al., 2005*; *Dexter et al., 2017*). At a given site, it is possible to find between 45 to 60 species coexisting (*Valencia et al., 2004*; *Endara & Jaramillo, 2011*; *Endara et al., 2021*). For *Inga*, species identification can be challenging because genetic and morphological variation is low (*Richardson et al., 2001*; *Dexter, Pennington & Cunningham, 2010*; *Dick & Webb, 2012*). Phylogenetic information based on plastid and nuclear ribosomal DNA markers have failed to resolve phylogenetic relationships
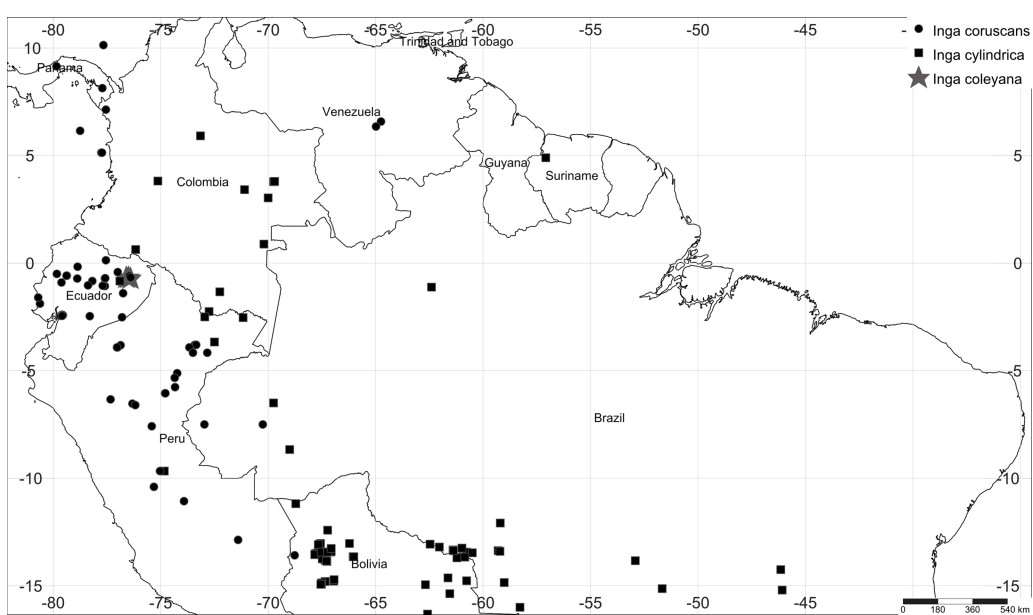

**Figure 1** **Map of collections of *Inga coleyana* M.J. Endara & J.E. Guevara sp. nov. and the most closely related species, *Inga cylindrica* (Vell.) Mart. and *Inga coruscans* Humb. & Bonpl. ex Willd. in the Amazon basin.** Black solid circles represent specimens' locations for *I. coruscans*, black solid squares for *I. cylindrica* and grey solid stars for *I. coleyana*.

within the genus *Inga* (*Richardson et al., 2001*; *Dexter, Pennington & Cunningham, 2010*), probably as a result of recent and rapid diversification (*Richardson et al., 2001*). Recent phylogenetic evidence based on targeted enrichment of 810 nuclear genes (*Nicholls et al., 2015*; *Koenen et al., 2020*; *Endara et al., 2021*) resolves species-level relationships within the genus. Moreover, this new phylogenetic reconstruction suggests that the taxon herein described represents a different, as yet unidentified species, closely related to *Inga cylindrica* and *Inga coruscans*, both widely distributed across the Amazon basin (Fig. 1).

Over the last few years, we have been studying a 90 one-hectare plot network strategically distributed throughout the Ecuadorian Amazon and a 50-hectare plot located in Yasuní National Park, representing in total 140 ha of lowland Amazonian forests that have been sampled. This network covers most habitats in the Amazon, including terra firme, white sand, varzea and igapó forests (*Valencia et al., 2004*; *Guevara Andino et al., 2017*; *Guevara Andino et al., 2021*). In these plots, we have leveraged recent advances in metabolomics and have performed an unprecedented analysis of the plant chemical diversity. After taking into consideration morphological, chemical, and phylogenetic analyses, as well as data from our plot network, we conclude that the species herein described represents a new rare and probably endemic taxon in the hyper-diverse genus *Inga*. Thus, the combination of plot data, phylogenetics and chemocoding can accelerate the pace of species recognition in Amazonian forests.

## MATERIALS & METHODS

The electronic version of this article in Portable Document Format (PDF) will represent a published work according to the International Code of Nomenclature for algae, fungi, and plants (ICN), and hence the final new names contained in the electronic version are effectively published under that Code from the electronic edition alone. In addition, new names contained in this work which have been issued with identifiers by IPNI will eventually be made available to the Global Names Index. The IPNI LSIDs can be resolved and the associated information viewed through any standard web browser by appending the LSID contained in this publication to the prefix "http://ipni.org/". The online version of this work is archived and available from the following digital repositories: PeerJ, PubMed Central SCIE, and CLOCKSS

We have inventoried a 90 one-hectare plot network across the Ecuadorian Amazon that covers most habitats in the Amazon, including terra firme, white sand, varzea and igapó forests (Fig. S1). In addition, our plot network spans a gradient in precipitation, temperature and soils fertility (*Guevara Andino et al., 2017*). In each of these plots we tagged and identified all individuals with a dbh $\geq$ 10 cm. We also made an exhaustive search of fertile and infertile specimens that could be assigned to the new species deposited in Herbario Nacional del Ecuador (QCNE), Herbario de la Universidad Católica del Ecuador (QCA) and Herbario Alfredo Paredes de la Universidad Central del Ecuador (QAP). We also reviewed specimen images of the genus *Inga* available at Instituto de Pesquisas da Amazonia (INPA) and the Field Museum (F) virtual herbaria (abbreviation according to *Thiers, 2019*). For both identified and unidentified specimens, we collected information about leaf size, leaf morphology, branch characteristics, pubescence, venation patterns and reproductive characters. For the specimens deposited in Ecuadorian herbaria we took measures using a BP electronic digital caliper and observations of reproductive characters were made using a ZEISS SteREO Discovery.V12 stereoscope.

Demographic and phenological data from the 50-ha Yasuní Forest Dynamic Plot (YFDP, *Valencia et al., 2004*) was used to determine mortality, recruitment, growth (increase in diameter) and phenology of the new taxon described here. Demographic censuses were carried out in 1995, 2002 and 2007.

Young expanding and mature leaves from individuals from *Inga* species were collected for genetic and metabolomic analysis, between 2013 and 2015. A previously published phylogeny of 164 *Inga* accessions reconstructed using targeted enrichment (HybSeq) dataset of 810 genes was used to determine the closest relative of the new taxon we describe in this study (*Endara et al., 2021*).

We also performed untargeted metabolomics in combination with a molecular networking approach to determine species-level fingerprints, which is defined as chemocoding (*Endara et al., 2018*; *Endara et al., 2021*). We used chemocoding to discriminate between the new taxon and its closest relatives. Following methods in *Endara et al. (2018)* and *Endara et al. (2021)*, leaves from five samples from each taxon were dried in the field using silica at ambient temperature. Later, they were ground and extracted using (60:40 buffer:ACN). The extract was analyzed using an Acquity UPLC I-Class system

and a Xevo G2 QToF mass spectrometer equipped with LockSpray and an electrospray ionization source (Waters, Milford, MA, USA). Data was collected in negative ionization mode using both single MS and MS-MS (Fast DDA, more details in *Endara et al., 2021*). MS-MS spectra of hundreds of compounds per species were used to determine chemical similarity. We assigned compounds to the class level using our in-house combinatorial database as well as through spectral matching to MS-MS databases and in silico prediction (*Endara et al., 2021*).

Finally, in order to delimitate the geographic range of the new species we performed analyses of extent of occurrence (EOO) and area of occupancy (AOO) in the online tool GeoCAT (*Bachman et al., 2011*). Collection and research permits were granted by the Ministry of Environment of Ecuador, permit MAATE-DBI-CM-2021-0187.

# RESULTS

## Taxonomic treatment

*Inga coleyana* M.J. Endara & J.E. Guevara, *sp. nov.*

Similar to *Inga cylindrica* (Vell.) Mart. in presenting a spike inflorescence but differing from this species by its larger (7.1–10.1 × 2.2–3.9 cm in *I. coleyana*) leaflets, non-canaliculate rachis, three to five pair of leaflets in *I. cylindrica vs.* three pairs of leaflets in *I. coleyana*, cyathiform foliar nectaries between leaflet pairs, sessile to slightly stipitate, bracts at the base of the inflorescence short and caducous, expanded, lax spike with up to 40 small pedicellate flowers, larger corolla tube (4–5.2 mm *vs.* 3.5–4.9 mm in *I. cylindrica*), puberulent calyx tube, two to four inner corolla lobes covering the staminal tube and non-reflexed corolla tube lobes. Also, morphologically similar to *Inga coruscans* Humb. & Bonpl. ex Willd., but differing by having brochidodromous venation *vs.* eucamptodromous venation, assymetrical leaf base *vs.* narrowly attenuate or acute leaf base in *I. coruscans* and by lacking a very congested spike inflorescence *vs.* lax spike inflorescence in *I. coleyana*.

Type: Orellana, Parque Nacional Yasuní-ECY, Parcela de 50 ha-PDBY. Bosque siempreverde de tierras bajas del Napo-Curaray (BsTa02), áreas de tierra firme colinado y de bajíos. Especies representativas: *Parkia nitida*, *Iryanthera hostmannii*, *Rinorea lindeniana*, *Iriartea deltoidea*. Árbol de 20 m, flores blancas. 03°38′S y 76°30′W, Alt: 200–300 m, 19 de agosto 2015, fl., *A.J. Pérez & W. Loor 9236* (holotype: QCA 236638!).

Type: Orellana, Yasuni National Park-YNP, YFDP 50-ha plot. Napo-Curaray lowland terra firme forests, hilly terrain with interspersed valleys. Common tree species: *Parkia nitida*, *Iryanthera hostmannii*, *Rinorea lindeniana*, *Iriartea deltoidea*. Subcanopy tree 20 m height, white flowers. 03°38′S y 76°30′W, 200–300 masl, August 19th 2015, fl., *A.J. Pérez & W. Loor 9236* (holotype: QCA 236638!).

Canopy trees up to 20–25 m tall; trunks cylindrical, external bark yellowish; branchlets glabrous, cylindrical, terete. Stipules 1–2 mm long, linear, caducous. Leaves alternate, paripinnate, with three pairs of leaflets; petioles 1.2–5.7 cm long, pulvinus (0.3–)0.4–1.2(−1.3) cm long, unwinged, terete and glabrous; foliar rachis non-canaliculate, unwinged,

4.2–15.5 cm long, 0.1–0.2 cm width; foliar nectaries cyathiform, sessile when mature or slightly stipitate when immature 1.3–1.9 mm width; young expanding leaves pale pink; leaflets (4.1–)7.1–10.1(–14) × (1.2–)2.2–3.9(–5.2) cm, papiraceous, subsessile, elliptical, apex attenuate, base obtuse or slightly asymmetrical, glabrous; secondary venation brochidodromous, prominent in the abaxial surface, slightly prominent in the adaxial surface, anastomosed 2–3 mm from the margin, green pale lustrous when live, (5) six to nine pairs of secondary veins, ascending and forming an angle of 45°, tertiary venation reticulate. Inflorescence axillary, groups of three to five expanded, lax, spikes, 30–40 flowered; peduncule (1.3–)1.7–3.9 cm long, slender, glabrous, without bracts. Flowers sub-sessile, pedicel 0.20–0.61 mm long, flowers white when mature, 10.2–12 mm long (including exerted portion of the stamen); calyx tube green, 1–2 mm long, 1–1.4 mm wide, glabrous, cyathiform, slightly expanded from base to apex, six triangular lobes, apex cuspidate; corolla tube campanulate 4–5.2 mm long, 2.2–2.7 mm wide, greenish yellow, glabrous and glossy, five to six triangular lobes, (1.05–)1.15–1.95 mm long, (0.8–)0.9–1.2 mm wide, two to four inner lobes covering the staminal tube, 3.7–4.2 mm long, 0.6–0.85 mm wide; stamens 43–55(–60) per flower, staminal tube white, 4.4–4.95 mm long and 0.6–1.2 mm wide, exerted (0.4–) 0.5–0.9(–1.2) mm from the corolla, free filaments (3.1–)5–6.35 (–7.1) mm long; ovary 1.5–1.7 mm long, 1-carpellate, glabrous, style not exceeding the staminal filaments, stigma head truncate, ovules ca. 8–16. Fruit a slender, long and flat pod, 14–15.5 × 2.4–2.8 cm with obtuse apex, straight, margins slightly expanded at maturity, valves face slightly convex around seeds, the sutures subligneous and slightly undulating around seeds, glabrous. Mature seeds not seen. Figures 2–3.

*Additional Specimens Examined*—**Ecuador**: Orellana, Parque Nacional Yasuní-ECY, vía NPF-Tivacuno, km6 +800. Bosque húmedo tropical, áreas de tierra firme colinado. Especies representativas: *C. sciadophylla*, *Bellucia pentamera*, *Ochroma pyramidale*, *Piper aduncum*, *Iriartea deltoidea*. Árbol de 25 m, −0,68222 S, −76,4072222W, 200-300 m, 31 de agosto 2009, (fl), *A.J. Pérez y W. Santillán* 4316 (QCA, MO); Orellana, Parque Nacional Yasuní-ECY, Parcela de 50 ha-PDBY. Bosque húmedo tropical, áreas de tierra firme colinado y de bajíos. Especies representativas: *Parkia nitida*, *Iryanthera hostmannii*, *Rinorea lindeniana*, *Iriartea deltoidea*. Árbol 25 m, vainas verdes. −0,633333 S, −76,5 W, 200–300 m, 26 marzo 2008, (fr), *A.J. Pérez y P. Alvia* 3949 (QCA, MO);

*Additional Specimens Examined*—**Ecuador**: Orellana, Yasuni National Park YNP, km6 +800 NPF-Tivacuno road. Tropical lowland terra firme forests, hilly terrain. Common tree species : *C. sciadophylla*, *Bellucia pentamera*, *Ochroma pyramidale*, *Piper aduncum*, *Iriartea deltoidea*. Tree 25 m height, −0,68222 S, −76,4072222W, 200–300 m, August 31st 2009 (fl), *A.J. Pérez y W. Santillán* 4316 (QCA, MO); Orellana, Yasuni National Park YNP, YFDP 50-ha plot. Tropical lowland rainforests, hilly tearrain with interspersed valleys. Common tree species: *Parkia nitida*, *Iryanthera hostmannii*, *Rinorea lindeniana*, *Iriartea deltoidea*. Tree 25 m height, mature pods green coloured. −0,633333 S, −76,5 W, 200–300 m, March 26th 2008, (fr), *A.J. Pérez y P. Alvia* 3949 (QCA, MO).

**Etymology**—We named the new species in honor of Phyllis D. Coley an evolutionary biologist, renowned for her important contributions to the understanding of the ecology and evolution of the interactions between plants and herbivores in tropical rainforest. For

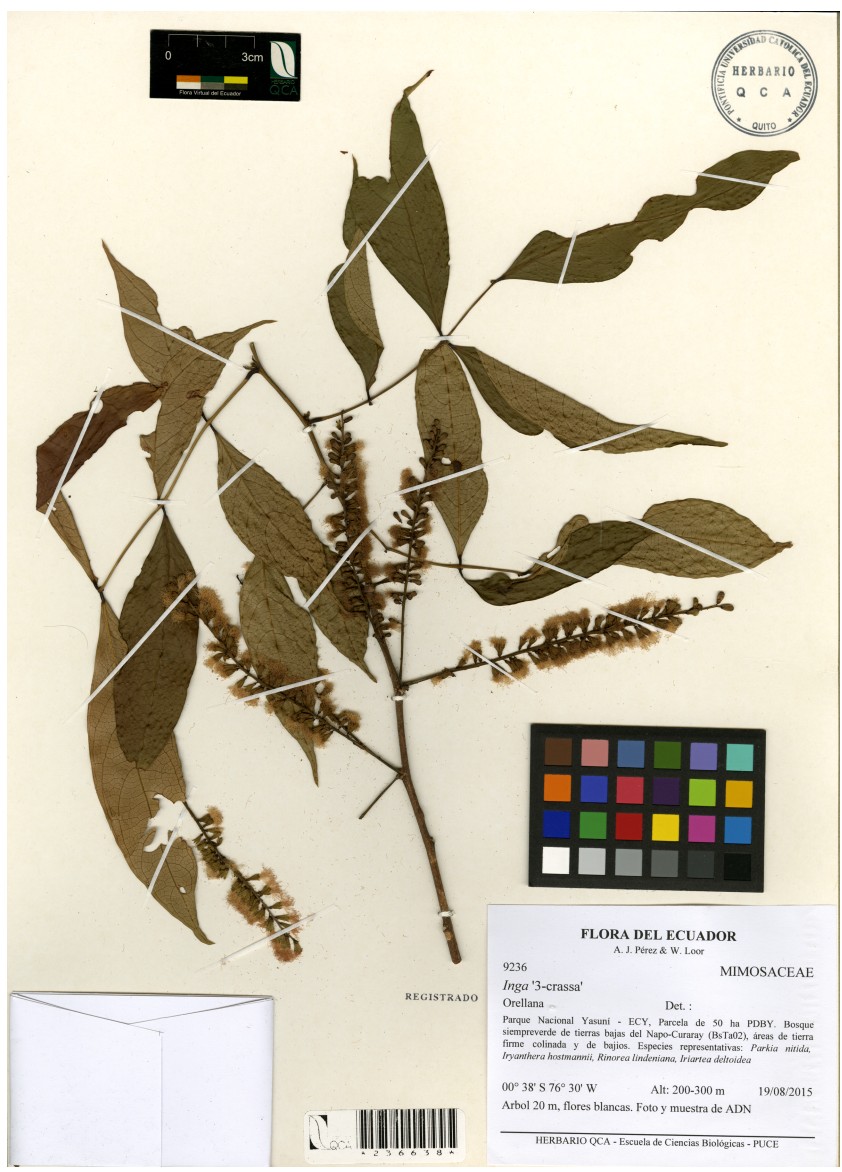

**Figure 2** *Inga coleyana* **M.J. Endara & J.E. Guevara sp. nov.** Image from the holotype at QCA (*A.J. Pérez & W. Loor* 9236).

more than 25 years, she has been working on the genus *Inga*, which has become a model system thanks to her contributions.

**Chemocoding analysis**—Metabolomic analyses of *Inga coleyana* and its closest relatives, *I. cylindrica* and *I. coruscans* (Fig. 4), suggest differences between species. The molecular network based on MS-MS spectra, shows divergent chemistry between the three species (Fig. 5A), with *I. coleyana* presenting 182 unique compounds (Fig. 5B). *Inga coleyana* chemistry is based on saponins, whereas *I. cylindrica* and *I. coruscans* produce a series of dihydroflavonols in addition to saponins.

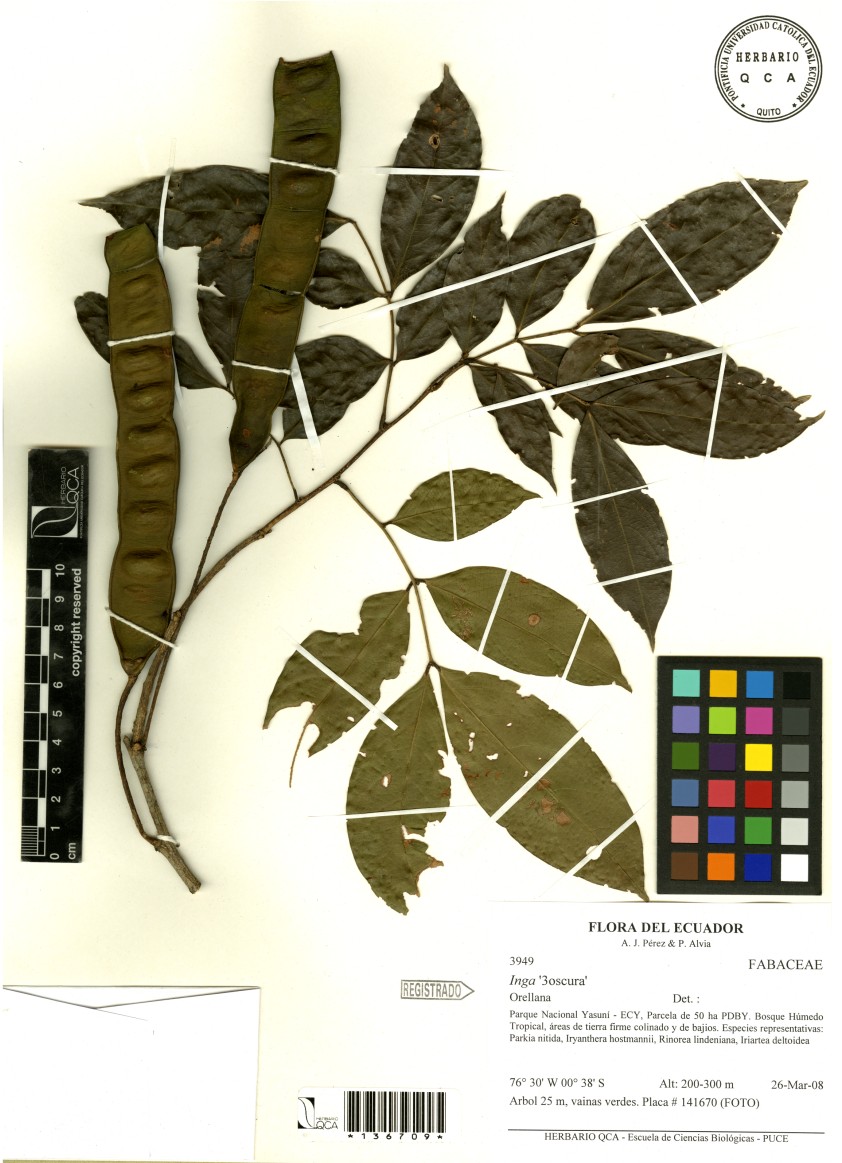

**Figure 3** *Inga coleyana* **M.J. Endara & J.E. Guevara sp. nov.** Image from the paratype at QCA (*A.J. Pérez & P. Alvia* 3949).

*Ecology and Distribution*—*Inga coleyana* is a small-medium sized tree that inhabits the terra firme forests below 400 masl on fertile soils in the Ecuadorian Amazon. *Inga coleyana* shows a habitat association with flat surfaces at the top of hills with well-drained soils and high clay content, as suggested by its local spatial distribution within the YFDP located at the Yasuní National Park, in Ecuador (Fig. 6A). In this plot, 30 individuals have been recorded, seven with dbh ≥ 10 cm, and 23 with dbh ≤ 10 cm. Only three individuals with dbh ≥ 30 cm have been found in this plot. The annual mortality rate of this species was 1.98% between 1995 and 2002 and 4.19% between 2002 and 2007. The rate of recruitment
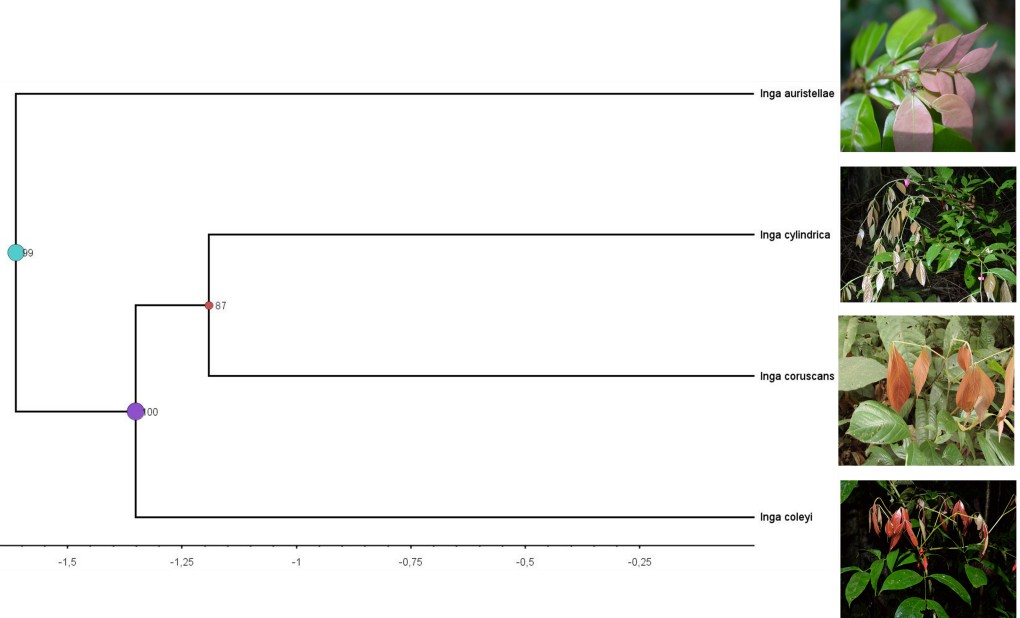

**Figure 4** Clade containing the most closely related lineages of *Inga coleyana* M.J. Endara & 442 J.E. Guevara sp. nov. This clade was adapted from a resolved phylogenetic tree based on 165 accessions of Inga across its distribution (*Endara et al., 2021*). Numbers in black represent posterior probabilities values for node support. Photos show juvenile individuals with fully expanded and young leaves.

was 2.00% between 1995 and 2002 and 2.9% between 2002 and 2007. There was no significant difference in growth rate between 1995–2002 (4.707 mm per year) compared with the period 2002–2007 (4.701 mm per year).

At regional scale and according to our tree species abundance data from the 90 one-hectare plot network, *Inga coleyana* appears to be relatively rare and possibly endemic to the Ecuadorian Amazon. The only two known populations of *Inga coleyana* are located inside the Yasuní National Park. Some of the most of conspicuous floristic elements of these forests include the families Fabaceae, Moraceae, Myrtaceae, and Sapotaceae both dominant and diverse. In this region the species ranks last in median abundance among all *Inga* species, with less than 1 individuals ≥10 cm dbh/ha. Also, according to published data and online specimens deposited at Field Museum virtual Herbarium no records of *I. coleyana* have been made in the adjacent Peruvian department of Loreto or the Putumayo-Caquetá region in Colombia (*Guevara Andino et al., 2017*; *Pitman et al., 2014*; *Pitman, 2004*).

*Phenology*—Flowering occurs from August to December; fruiting occurs between March and May (Fig. 6B).

*Conservation Status*—*Inga coleyana* is known from only two populations inside the Yasuní National Park in the Ecuadorian Amazon. The first corresponds to a small population represented by the type collection in the 50-ha YFDP. In this plot, 30 individuals have been reported, with only three of them as adults with a diameter at breast high ≥ 30 cm. The second one corresponds to a population in the surroundings of km 6 of the NPF-Tivacuno road inside the Yasuní National Park. The estimates of extent of occurrence
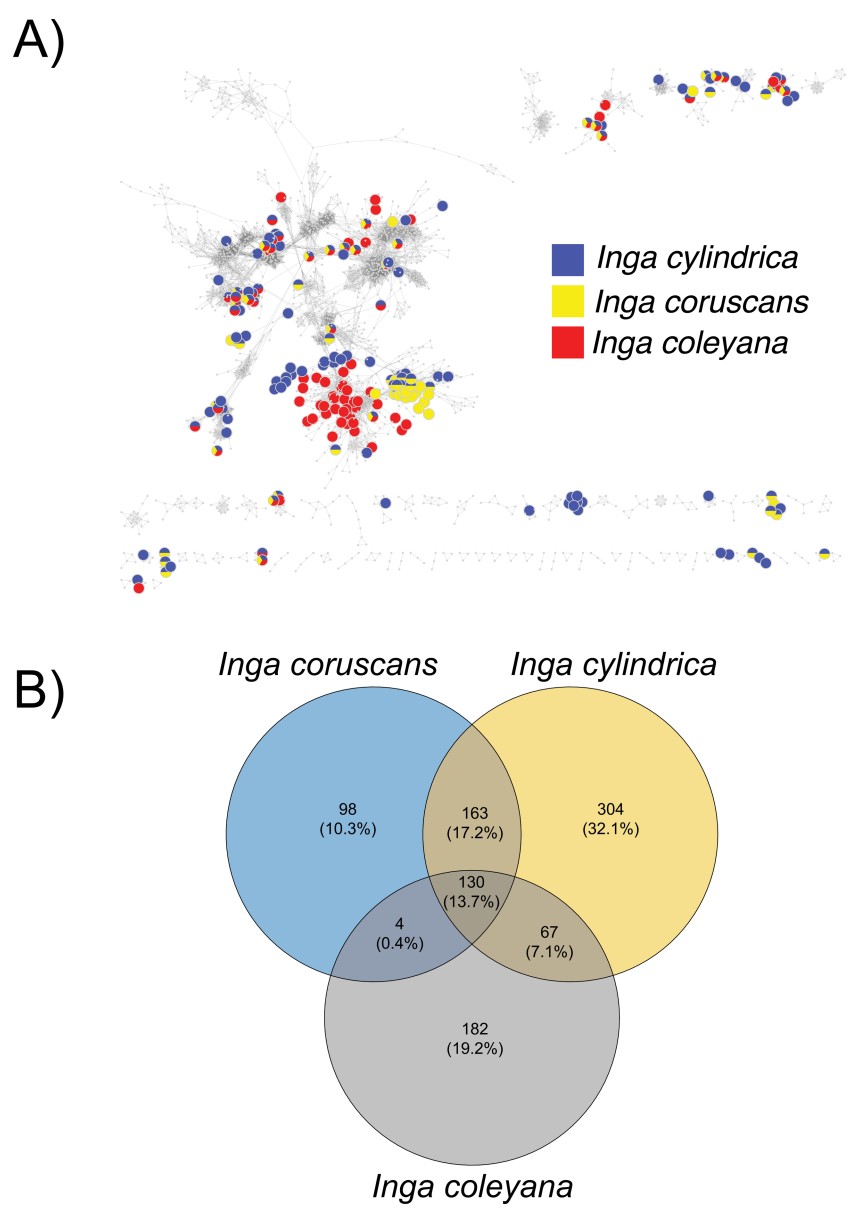

**Figure 5** **Compound base molecular network.** (A) A similarity network of secondary metabolites isolated from 167 species of Inga. Each dot is a compound, and lines connect structurally similar compounds based on MS-MS spectra. In color, are shown compounds isolated from *Inga coleyana* M.J. Endara & J.E. Guevara and its sister species. (B) Venn diagram showing the degree of overlap of secondary metabolites between *I. coleyana* and its sister species. Number of compounds, and percentages in parentheses, that are shared between species are represented in the overlap between circles.

(EEO) are 288.287 km$^2$, and for area of occupancy (AOO) are 16 km$^2$ (*Bachman et al., 2011*). Although the two populations of *I. coleyana* are formally protected inside the Yasuní National Park, its rate of recruitment and mortality as well as its small EEO and AOO together with the rate of deforestation in adjacent areas to the Yasuní National Park, support the assessment of *Inga coleyana* as Vulnerable (VU D2), following criteria from

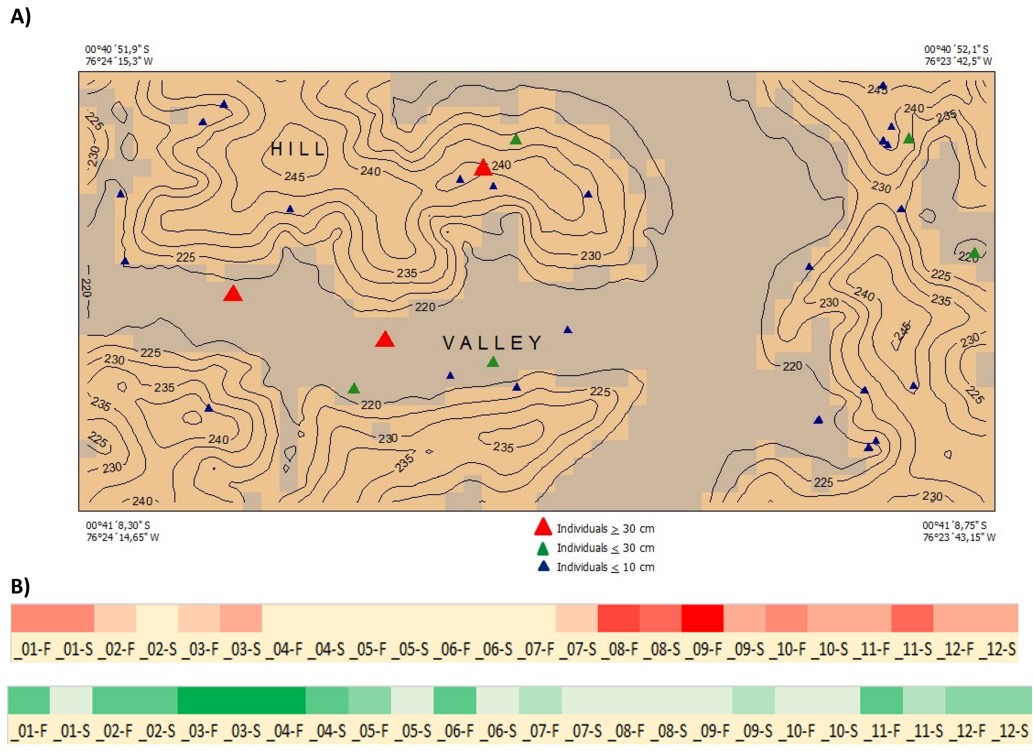

**Figure 6** **Distribution and phenology of *Inga coleyana* M.J. Endara & J.E. Guevara sp. nov.** (A) Map of individuals of *I. coleyana* in a 50-ha plot at Yasuní National Park, Ecuador. Three size classes are displayed: ≥1–10 cm (blue triangles), ≥10–30 cm (green triangles), and ≥30 cm dbh (red triangles). Topographic contours are at 5-m intervals, and the coordinates of each corner of the plot are given. (B) Flowering and fructification times in I. coleyana; numbers represent each month of the year, F, first and S, second monitoring period for each month. The monitoring of fruits and flowers traps were done every 15 days.

*IUCN (2014)*. However, extensive clear-cutting of the Amazon lowland forests along the northern portion of the Ecuadorian Amazon might threaten in the future more unknown populations of this species in this region.

## DISCUSSION

Phylogenetic analysis including 165 accessions of *Inga* across its distribution suggests that the sister lineages for *Inga coleyana* are *Inga cylindrica* and *Inga coruscans* (Fig. 4), both known to occur in Central, Western Amazonia and the Guiana Shield. The new taxon can be differentiated from these species by having shorter leaflets, cyathiform nectaries, glabrous leaflets with attenuate apex and obtuse base, flowers born in a lax spike, small flowers with short campanulate corolla and inner lobes enclosing the staminal tube. *Inga coleyana* also differs from *I. cylindrica* by having three pairs of leaflets (*vs.* three to five pairs of leaflets in *I.* cylindrica) and a larger ovary (4–5.2 *vs.* 3.5–4.9 mm in *I. cylindrica*) (Table 1). *Inga coleyana* differs from *I. coruscans,* a tree species that occurs on inundated forests (*e.g.,* varzea forests) in Brazil, Colombia, Ecuador and Peru, by having more leaflets (two to three pairs of leaflets for *I. coruscans*), with a base narrowly attenuate or acute,

**Table 1** Diagnostic characters for *Inga coleyana* (Leguminosae, Mimosoid clade) and its closest relatives as well as their geographical distribution in the Amazon basin: Northwestern Amazon (NWA), Southwestern Amazonia (SWA), Central Amazon (CA) and the Guiana Shield (GS).

| Characters | *I. coleyana* | *I. cylindrica* | *I. coruscans* |
|---|---|---|---|
| Leaflets pairs | 3 | 3–5 | 2–3 |
| Leaflets apex | Attenuate | Acute to attenuate | Shortly-and arrowly attenuate |
| Leaflets base | Obtuse | Cuneate | Narrowly attenuate or acute |
| Leaflet venation | Brochidodromous | Eucamptodromous to brochidodromous | Eucamptodromous |
| Rachis | Terete or slightly canaliculate | Strongly canaliculate | Semiterete |
| Foliar nectary | Sessile and cyathiform | Sessile and patelliform | Sessile patelliform or cyathiform |
| Inflorescence structure | Lax spike | Congested spike | Very congested spike |
| Corolla tube (length) | 4–5.2 mm | c. 3 mm | 4–5 mm |
| Corolla tube lobes | 5-6 glabrous and glossy | 5 glabrous | 4 puberulous |
| Calyx lobes (length) | 0.2–0.5 mm | 0.25 mm | 0.5 mm |
| Staminal tube (length) | 5.2–7.8 mm | 6–9 mm | 6.5–7 mm |
| Staminal tube exserting corolla tube | 1.05–1.95 mm | c. 3 mm | 3–3.5 mm |
| Stigma morphology | Capitate | Simple | Simple |
| Geographic distribution | Ecuador only | NWA, CA and GS | NWA, SWA |

glabrous terete rachis, very congested spikes with shorter peduncles (0.8–2.5 mm *vs.* 1.3–3.9 mm in *I. coleyana*) and a simple stigma. In addition, untargeted metabolomic data shows that *I. coleyana* express a unique defensive chemistry, different from its closest relatives. Together, morphological, phylogenetic and chemical evidence suggest that *Inga coleyana* is a novel taxon well differentiated from other congeneric species inside the Neotropical genus of trees *Inga*.

Among the few specimens of *Inga coleyana* that were recovered in Ecuadorian herbaria, one of them was wrongly assigned to *Inga coruscans*. The rest of the studied specimens were unidentified. The case of *I. coleyana* is one of the many examples of new species that remains undiscovered due to poor sampling and lack of fertile material deposited in herbaria. This can be solved, as we demonstrated here, with the combination of long-term plot monitoring and novel techniques for species identification such as chemocoding. Long-term plot monitoring allows a more rapid collection of fertile specimens for tree species that occur at low density and frequency in Amazonian forests. This is particularly important for species description because ecologists can revisit these plots, increasing the probability of collecting specimens with flowers and fruits. Since *I. coleyana* was identified in 2015 as a new species, almost six years passed until fertile specimens for taxonomic description were collected. This lapse of time is considerably less than the average time that takes species descriptions, which can be of decades (*Baker et al., 2017*; *Ter Steege et al., 2019*). The advent of omics for large-scale plant identification, such as metabolomics and genomics, provides a valuable additional source of data that can facilitate the identification of plant species, with the additional benefit of providing information on functional traits and evolutionary history for many groups of trees (*Pirie, Chatrou & Maas, 2018*; *Guevara Andino et al., 2019*; *Endara et al., 2018*; *Endara et al., 2021*). Thus, accelerating taxonomic recognition in challenging tropical biomes would benefit from

the synergy produced by these different approaches. This is particularly important in the current scenario of habitat loss, fragmentation and deforestation that Amazonian forests are suffering.

## CONCLUSIONS

We have show here that the integration of long-term plot monitoring and novel techniques such as metabolomics and phylogenomics can improve our ability to delimitate and accelerate the pace of species discovery in species-rich groups of Amazonian trees. In our study, we have demonstrated that chemical fingerprinting in combination with demographic, morphological and phylogenetic data from long-term plot monitoring could help to a better diagnosis of rare species that might be new to science. This is particularly important for species recognition in hyperdiverse groups such as *Inga* where cryptic diversity makes difficult to delimitate and differentiate new species. Moreover, we propose that chemocoding can be especially amenable to unfold species diversity when hundreds or thousands of samples need to be analyzed in challenging biomes such as Amazonian tree communities.

## ACKNOWLEDGEMENTS

We thank the staff at the herbaria HUTI, QCA and QCNE for their support and for facilitating loans of material that made this study possible.

### Funding

This work was supported by the Universidad de las Américas research grant (FGE.JGA.20.04) and the Artificial Intelligence for Species Discovery National Geographic Grant (NGS-72018T-20). The funders had no role in study design, data collection and analysis, decision to publish, or preparation of the manuscript.

### Grant Disclosures

The following grant information was disclosed by the authors:
Universidad de las Américas research grant: FGE.JGA.20.04.
Artificial Intelligence for Species Discovery National Geographic Grant: NGS-72018T-20.

### Competing Interests

The authors declare there are no competing interests.

### Author Contributions

- Juan Ernesto Guevara Andino conceived and designed the experiments, performed the experiments, analyzed the data, prepared figures and/or tables, authored or reviewed drafts of the article, and approved the final draft.
- Consuelo Hernández analyzed the data, authored or reviewed drafts of the article, and approved the final draft.

- Renato Valencia analyzed the data, authored or reviewed drafts of the article, and approved the final draft.
- Dale Forrister analyzed the data, prepared figures and/or tables, and approved the final draft.
- María-José Endara conceived and designed the experiments, performed the experiments, analyzed the data, authored or reviewed drafts of the article, and approved the final draft.

## Field Study Permissions

The following information was supplied relating to field study approvals (i.e., approving body and any reference numbers):

Collection and research permits were granted by the Ministry of Environment of Ecuador (MAATE-DBI-CM-2021-0187).

## Data Availability

The dhemical data and analysis for chemical similarity is available at GitHub: https://github.com/dlforrister/Forrister_et_al_2022_evolution_of_inga_chemistry.

## New Species Registration

The following information was supplied regarding the registration of a newly described species:

Inga coleyana LSID: 77301122-1

## Supplemental Information

Supplemental information for this article can be found online at http://dx.doi.org/10.7717/peerj.13767#supplemental-information.

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
