# Peer review of "Accelerating the discovery of rare tree species in Amazonian forests: integrating long monitoring tree plot data with metabolomics and phylogenetics for the description of a new species in the hyperdiverse genus Inga Mill"

_PeerJ, doi:10.7717/peerj.13767_

## Round 0.1 · original submission · Minor Revisions

I am pleased to say the reviewers and I are in agreement that the manuscript is publishable with minor modifications. Please read the feedback carefully as there are many constructive suggestions.

Reviewer 1 ·

Basic reporting

This manuscript provides strong evidence for supporting the recognition of the new taxon, complemented with demographic data rarely incorporated in other studies outside of a protected site. However, the epithet of the new name needs to be changed.
The protologue emphasizes morphological contrasting characters, and the Table 1 provides and complement those characters with closely related species.
Untargeted metabolomics has shown to be a consolidate valuable method for complementing species recognition. This manuscript proofs the validity of this method, and I found that its most valuable contribution is to show the importance and benefits of long-term study sites within a protected area related to understanding biodiversity composition. The recognition of species novelties is important not only for “speeding the pace of species discovery”, but also because it demonstrates the complexity of biological systems like those found in the Amazonian basin as previously published by the authors.
The introduction, however, needs changes. It should incorporate evidence hinted in previous phylogenetic analysis about taxa not formally recognized.

Experimental design

The use of all available data is supported and brings a taxonomic decision in a structured manner.
Metabolomics appears to be an effective method in Inga, as previously demonstrated by one of the authors in proving the value of those chemocodes. Some of these methods could be costly, and maybe advantages of metabolomics should be mentioned in this regard.
Their phylogenetic support should be clearly presented in the introduction and not limited to the discussion, since it has guided the affinities of the new taxon to other species in the genus. The supplement incorporates the tree not published or included as supplement in the cited paper by Endara et al. 2021. Since it is added in the supplement of this manuscript, it needs a legend clarifying the clade in green, values of support, and the use of codes within the tree.

Validity of the findings

Their conclusion in recognizing a new species is valid, well-supported and presented. However, there is a correction needed for the new epithet name.
Line 147 requires a change that impacts citations of the name for the whole manuscript. Because the epithet honors a female researcher, to comply with Art. 60.8c of the ICN the epithet should be spelled “coleyana” and not coleyi.

Additional comments

First paragraph requires changes for attributing exploration to the discovery of the Amazonian tree flora. Amazonian people were using those plants well before contact with Western naturalists. Line 52 mentions the challenge of finding new species, it should instead be recognition, especially as in this case of cryptic species within a speciose genus, as stated in line 73.
The use of the term “discovery” and “finding” should be changed to recognition. Since, as in the case of the new taxon it was vegetatively already hinted in the field name for the collected sites.
The availability of long-term monitoring of trees has allowed the confirmation of this new taxon. However, it is irrelevant for the suggestion in lines 284–285 that the lapse of time is less than average time for a formal description. There are so many factors for a new taxon not being described, and it could have happened to this species that fortunately have an specialist for the genus, and methodology already set for it.
Line 91 should include the word final, since ICN requires effective publication according to Arts. 29.1 and 30.2
Line 63 should add other molecular tools.
Line 101–104 clarify “In each of these plots..”, since line 226 mentions 90 ha-plots.
Line 121, delete “published” and replace for “developed”
Line 165, as a suggestion indicate the Unique number (barcode) of QCA
Lines 245–246. Directly indicate the estimates of occurrence and occupancy, since the use of GeoCat is already mentioned in methods
Lines 263, and 265. The genus name should not be abbreviated at the beginning of a sentence
Minor observations
Lines 164, 167, 177. Change to en-dash
Line 185, add space after “exerted”
Line 188. Change to multiplication sign (as in line 172)

Reviewer 2 ·

Basic reporting

Review of the manuscript titled: Accelerating the discovery of rare tree species in Amazonian forests: Integrating long monitoring tree plot data with metabolomics for the description of a new species in the hyperdiverse genus Inga Mill. by Juan Ernesto Guevara Andino and collaborators.

The authors present a thorough and well-illustrated study of three closely realted taxa in the speciose, newtropical tree-genus Inga. Species delimitation in Inga is notoriously difficult bsed on morphology alone, and novel approaches are thus constantly being sought.

Guevara et al. apply interesting approach to species delimitation, in addition to morphology vegetative and fertile characteristics they include matabolomics and phylogenomics, which may prove useful in closely related rainforest species, beyond the genus Inga.

Experimental design

The design and experiment are interesting, novel and make use of the wealth of molecular data and plot-data available on the genus Inga.

Validity of the findings

Valid and new to science.

Additional comments

I have made some minor comments on the uploaded Word document, which need to be addressed prior to publication.

Annotated reviews are not available for download in order to protect the identity of reviewers who chose to remain anonymous.

---

## Round 0.2 · accepted · Accept

I am satisfied you have made every reasonable effort to incorportae the feedback from the reviewers. I have read through the manuscript and am hapy you have made changes that meet the spirit intended in the reviews.